# The Impact of Delivering School-Based Wellness Programs for Emerging Adult Facilitators—A Quasi-Controlled Clinical Trial

**DOI:** 10.3390/ijerph19074278

**Published:** 2022-04-02

**Authors:** Moria Golan, Dana Tzabari, Maya Mozeikov

**Affiliations:** 1Department of Nutritional Sciences, Tel-Hai College, Upper Galilee 1220800, Israel; danatzabari91@gmail.com (D.T.); mayamozeikov@gmail.com (M.M.); 2Shahaf, Community Based Facility for Body Image and Eating, Ganey Hadar 7683000, Israel

**Keywords:** emerging adulthood, program’s facilitators, prevention program, personal attributes, academic supervision course

## Abstract

A quasi-controlled clinical trial included a university-based supervision course for facilitators of an interactive wellness school-based program. The study aimed to investigate how students that facilitate prevention programs are personally affected by delivering content related to self-esteem, body-image, and media literacy. In total, 66 university students who were either facilitators of preventive programs (intervention group) or non-facilitators (comparison group) completed questionnaires before, after, and three months following the program’s termination. All methods were performed following the Declaration of Helsinki regulations and Consort 2010 guidelines. Participants in the facilitator group demonstrated statistically significant superiority, with large effect size, regarding improvement in identifying advertisement strategies. Weight-related body-esteem, and the reduced impact of media messages’ pressure also had statistically significant superiority, with small effect size. The number of participants with pathological EAT-26 scores (>20) decreased from 5 to 2 in the facilitator group compared to an increase from 5 to 6 (no statistical significance) in the comparison group. Both groups demonstrated statistically significant decreases in eating disorder perceptions and behaviors from baseline to 3-month follow-up. Delivering a prevention program proved beneficial to facilitators, in addition to the target school pupils, and thus may be considered as part of the prevention programs’ effectiveness assessment.

## 1. Introduction

Emerging adulthood is a distinct period in which students rely more on themselves and challenge independent life while developing qualities for becoming self-sufficient. They engage in mature coping styles and relationships and assume adult responsibilities [1,2]. Social pressure, novelty-seeking, and high-intensity emotions may challenge students and frequently they are tempted to impaired self-care, disordered eating [3], substance abuse [4], and other unfavorable behaviors [5,6]. It is, therefore, interesting to investigate how students that facilitate prevention programs, are personally affected by delivering content related to self-esteem, body image, and media literacy. Although there is extensive literature regarding the impact of prevention programs on target populations [7,8,9,10], there are only a few publications discussing how facilitating a preventive program influence facilitators’ personal attribute.

Benefits of community engagement, such as peer-teaching, peer-support leaders, clinical internship, mentoring, and professional supervision, are well documented [11]. Service participation of social work undergraduate students, compared to nonparticipation, was associated with an increase in social self-confidence and leadership ability, understanding community problems and gaining interpersonal skills, conflict resolution skills, and ability to think critically [12]. In medical studies, similar benefits were reported by peer-teachers. Subject competency, pedagogical content knowledge, and deep self-observation skills were shown to impact both audience and facilitator [13]. Crenshaw [14], an undergraduate student peer facilitator in the “Team Up for Healthy Living”—a school-based obesity prevention project—reported that delivering the program resulted in changes in her diet and physical activity behaviors and improvements in academic/professional skills [14]. Jennings et al. [15] compared facilitators of a school-based prevention program focused on preventing unplanned pregnancies and sexually transmitted diseases with non-facilitators. Results showed facilitators reporting more personal opportunities to practice skills related to reducing the risk of contracting sexually transmitted diseases, more opportunities to talk to friends, parents, and sexual partners, more knowledge about sexual health problems, and being more able to refuse sexually significant actions. However, whether these facilitators applied their new knowledge and skills to real-life situations was not assessed [16]. Gilmartin et al. [17] reported in a systematic literature review on brief mindfulness practices for healthcare providers that none of the 14 studies reviewed found an effect on provider behaviors.

The reported studies have several limitations. Some lacked control groups [18,19], while others did not analyze the change in the personal application of knowledge or modeling [15], or its long-term impact [19,20]. Moreover, most information was qualitatively assessed [21]. Researchers have recommended further evaluation of personal changes that facilitators experience as they take on these leadership roles [17,22,23]. 

The current study evaluates how students that facilitate “Favoring Myself”, a school-based prevention program, are personally affected by delivering content related to self-esteem, body image, and media literacy. “Young Favoring Myself” is a manualized universal, interactive intervention with ten weekly, 90 min sessions [10]; topics are detailed in Table 1. The program includes a kit with detailed background material and a detailed guide for facilitators with structured session plans, including interactive activities to engage young adolescents (5th graders) verbally and non-verbally. To trigger situational interest, hands-on activities, novelty, surprise, and group work, were used. 

The research hypothesis was that facilitator’s self-esteem, self-care, media literacy, and body esteem will be improved following facilitating these topics in a school-based prevention program with a more prominent effect size than that observed during the same period among participants in the comparison group who did not facilitate this program. Both research arms’ participants were third-year college students in their emerging adulthood, facing challenging behavioral and health choices that directly impact their sense of well-being. To the best of our knowledge, no studies have reported the impact of preventive wellness programs on program facilitators’ wellness components—a gap that this paper addresses. 

The program’s topics target protecting factors (knowledge and understanding of self-care in various situations, self-esteem, positive body esteem, media literacy, positive defense mechanisms). The study hypothesis relied on Deci and Ryan’s self-determination theory [24], which assumes that when students achieve a sense of autonomy, relatedness, and competence through the support system provided while facilitating prevention programs and through the emphasis on certain topics and assignments, they are highly motivated and personally impacted. 

## 2. Materials and Methods

### 2.1. Design and Sample Recruitment

A controlled trial was designed to evaluate the impact of facilitating a school-based wellness program for facilitating students’ self-esteem, body-esteem, media literacy, and eating attitudes—topics that are the focus of the delivered prevention program. The change in these variables over the same period was compared in students that did not deliver this program. The research was performed within the real-world university setting and as such, randomization could not be performed. 

The facilitator group consisted of third-year college students, all of whom delivered the “Favoring Myself” program during 2019–2021. Applicants from the nutrition and education departments were interviewed by a group-dynamic expert who assessed their co-facilitation abilities, group leadership experience, and feedback mindset to be included in the facilitator group, subsequently paired to form co-facilitating pairs. 

The comparison group was recruited through the institute’s internal advertisement system. A public call was published in the internal college advertisement system calling for third-year students to fill out psychological tests that took approximately 20–30 min each. This was done three times during the course of six months and students were compensated with the Israeli shekel equivalent of USD 35. The eligibility criteria for the comparison group were similar for major studies, age, gender, and ethnicity. Inclusion criteria included having participated in a previous academic course, and exposing participants to all the topics presented in the intervention group’s program (Table 1), plus they had to have at least six months experience of group session or training delivery in any field. However, they never participated in delivering “Favoring Myself”, nor did they participate in the supervision group. By these inclusion criteria, we attempted to differentiate the intervention and comparison group by the ‘topics’ facilitation’ and by ‘belonging to the facilitators’ supervision group’.

### 2.2. Sample Size

The sample size was calculated using the EpiTools epidemiological calculation (Ausvet, Bruce, Australia), based on data from previous studies. The SDs of the Body-Esteem Scale total scores were 0.8 and 0.9. A significant difference was indicated by a change of 0.6 points in the average score, with a power of 80%, and α = 0.05. The calculation yielded a sample size of 27 pairs. Assuming a 10% expected dropout rate, the calculated sample size was 29 participants in each study arm.

### 2.3. Ethical Procedures

The study was approved by Tel-Hai College Institutional Review Board (No. 12/2018 and No. 8/2019). The pre-registered universal trial number is NCT03882242 (18/3/2019). All procedures and methods were performed following the Declaration of Helsinki regulations and Consort 2010 guidelines. Written informed consent was obtained from all participants. 

### 2.4. Intervention

“Young Favoring Myself” is an interactive program comprised of ten weekly, 90 min sessions on self-care behaviors, media literacy, self-esteem, and positive body image, topics which were mentioned as protective factors against risk behaviors [6,7,9]. It is an evidence-based intervention that was empirically supported and substantiated with research findings that demonstrate beneficial and predictable outcomes [10]. The program topics are described in Table 1. Facilitators used hands-on activities, novelty, surprise, and group work, incorporating age-tailored games into the sessions. The program is semi-structured, with flexibility that enabled facilitators to be creative while addressing their groups’ specific needs. 

### 2.5. Study Population

A total of 66 third-year college students (mean age 26.3) participated in the study. In total, 24 students (36%) were Education majors, and 42 students (64%) were Nutritional Sciences majors, with 33 participants in each study group (Figure 1). All eligible participants completed all assessments’ questionnaires. A total of 99% of the facilitators were women. The study population’s personal and sociodemographic characteristics are displayed in Table 2. No differences in age, gender, parental status, or socioeconomic status were found between the facilitator and the comparison groups. 

### 2.6. Facilitator Training

Program facilitators participated in 4 h workshops given on a weekly basis for 13 weeks (an entire college semester). The first author, who developed the “Young Favoring Myself” program curriculum, conducted the didactic supervision for the first two hours. In this platform, each pair of co-facilitators delivered to didactic supervision group one session simulation, as if it would be performed in the school classroom. The peer facilitators’ group served as the participating class, which required them to demonstrate self-observation and self-disclosure regarding their attitudes toward self-esteem, body-esteem, eating attitudes, and media literacy. The supervising professor ensured that session objectives were adequately addressed with age-appropriate pedagogic strategies. The peer facilitators and the professor provided feedback on the demonstration. A written reflection on the co-facilitator’s demonstration focused on strengths and scope for improvement. In the following 2 h, a social worker facilitated a group-dynamic activity to train the group on becoming successful prevention program facilitators. Each week, the session started with the previous week’s facilitation debriefing, discussing challenges and unique experiences.

### 2.7. Data Collection Procedure

A computerized self-administered questionnaire in Qualtrics XM software was completed by participants before and after program delivery, as well as three months following program termination.

### 2.8. Outcome Measures and Variables

Standardized instruments were used to measure the program’s effect on facilitators. All scales included in the study questionnaire which are validated Hebrew-translated versions. The study questionnaire included the following measures.

### 2.9. Personal and Demographic Information

Information regarding age, gender, parental and marital status, prior instruction/facilitation experience, and socioeconomic status was obtained.

### 2.10. Media Literacy and Pressures by Media

Media literacy was assessed using the Advertising Scale [23]. The scale contains one item that tests the identification of strategies used by advertisers and is reflected as a protective factor. It includes eight different strategies which participants choose from, while a higher number of strategies identified indicates higher media literacy. The strategies included: exaggeration, illusions, convincing using logic, convincing using emotions, romanization, idealization, realism, and intimidation. The “Pressures by Media” subscale of the Sociocultural Attitudes Toward Appearance Questionnaire-4 (SATAQ-4) was used to examine the internalization of media effects. This 4-item scale is rated on a 5-point scale. A higher average score indicates higher media pressure to change one’s appearance [25]. The Hebrew translation has been previously used in research [26]. In the current study, the subscales had high internal consistency, indicated by a mean Cronbach’s alpha value of 0.95.

### 2.11. Self-Esteem, Body-Esteem Scale, and Behaviors and Perceptions Associated with Eating Disorders

The 10-item Rosenberg Self Esteem Scale (RSE) [27] is ascendingly scored on a 4-point Likert scale. Higher scores suggest high self-esteem. The Hebrew version of the RSE has been previously validated in Israel [28]. In the current study, the subscales had high internal consistency, indicated by a mean Cronbach’s alpha value of 0.89. The Body Esteem Scale (BES) was used to assess body perceptions. The 23-item scale is divided into three subscales: Appearance, Weight, and Attribution. The mean total and subscale scores are rated on a 5-point Likert scale. Higher scores represent higher body esteem [29]. The Hebrew version of the BES has been used in previous research [10,30]. In the current study, the subscales had high internal consistency, indicated by a mean Cronbach’s alpha value of 0.96. The Eating Attitudes Test (Eat-26) was used to assess behaviors and attitudes associated with eating disorders. This scale is widely used as a screening tool for eating disorders. This 26-item scale is rated on a 6-point Likert scale. Instructions are to score the 6 response choices as 3, 2, 1, 0, 0, 0. The Hebrew translation of the EAT-26 [30,31] has been used widely in Israel for research and clinical purposes. The subscales had high internal consistency in the current study, indicated by a mean Cronbach’s alpha value of 0.83. All scales included in the study questionnaire were previously validated, Hebrew-translated versions.

### 2.12. Satisfaction Assessment

The post-intervention questionnaire consisted of a satisfaction assessment.

### 2.13. Statistical Analysis

SPSS (version 23, IBM Corp., Armonk, NY, USA) was used to perform formal statistical analyses, with significance considered at the *p* < 0.05. Variable’s normality was assessed using Kolmogorov–Smirnov and Shapiro–Wilk tests. Chi-squared tests for categorical variables and Wilcoxon Two-Sample Tests for continuous variables were conducted to test differences between intervention and comparison groups in the demographic variables. Independent sample *t*-tests (or Wilcoxon Two-Sample Tests) checked the baseline differences between the two study groups in the dependent variables. To account for significant differences between groups in baseline outcomes (Rosenberg Self-esteem scale and the Body-Esteem Appearance subscale), baseline scores were included in repeated-measures ANOVA analysis as covariates.

Normally distributed variables were tested using mixed model analysis. Non-parametric tests were used to analyze variables with non-normal distribution (ADVER- Advertising scale, EAT-26). Friedman tests were used to examine the differences between study times within each group, and Wilcoxon Two-Sample Tests were used to investigate differences between the two study groups at each time point. The effect size was Cohen’s d for independent sample T-tests, marginal R^2^ for the mixed models, Wilcoxon effect size for the Wilcoxon Two-Sample Tests, and Kendall’s W for the Friedman test.

## 3. Results

### 3.1. Baseline Differences in Outcome Measures between Facilitators and Comparisons

Differences between facilitators and comparison group in outcome measures at baseline are shown in Table 3 and Table 4. No differences were found between the intervention and the comparison groups regarding most of the study outcomes (Body Esteem-total score, Body Esteem-Weight subscale, SATAQ-4, Self-care, and Eat-26 mean score). Significant differences were found at baseline regarding the Rosenberg Self-Esteem scale (*p* = 0.021), Body Esteem appearance (*p* = 0.040) and attribution (*p* = 0.048) subscales. Thus, these baseline measures were included as covariates in the repeated-measures analysis performed to assess between-group differences.

Eat-26 mean baseline score was higher, although not statistically significant in the intervention group compared to the comparison group (small effect size). Still, both scores were under 20—the pathological threshold which serves as a screening score.

### 3.2. Differences in Outcome Measures between Facilitators and Comparisons along Assessment Times

A mixed-models analysis regarding the normally distributed variables was performed to assess the main effects of group and time and their interaction (Table 5). Participants in the intervention group achieved superior improvements in body esteem regarding weight [F (2128) = 3.28, *p* < 0.05, marginal R^2^ = 0.025] and media pressures filtration throughout assessment times [F (2128) = 3.28, *p* < 0.005, marginal R^2^ = 0.011], though with a small effect size. This was demonstrated by the significant group × time interactions in both outcomes (Body-Esteem Weight subscale and the SATAQ-4 media subscale).

Analyses did not detect a Group × Time interaction for the Rosenberg Self-esteem (RSE) scale, the Body-Esteem Appearance, Weight subscale (with or without standardization of baseline values as covariates) and Self-care. However, significant main Time effects were found for these outcomes, suggesting that both groups tended to achieve improvement in these outcomes along the time, although with small effect sizes.

Analysis for the two non-parametric variables (Advertisements’ Strategies and Eat-26) are shown in Table 6. Regarding the identification of advertisement strategies, a statistically significant superiority of the intervention group was found compared to the comparison group with large effect size at the program conclusion (*p* = 0.0005, W = 0.43) and follow-up assessments (*p* = 0.025, W = 0.275). Significant within-group differences were found between study times in both the comparison (*p* = 0.002, W = 0.38) and the intervention groups (*p* < 0.001, W = 0.57). Suggesting that both groups tended to achieve improvement in the ability to identify advertisement strategies, with moderate effect size (0.38) in the comparison group and large effect size (0.58) in the intervention group.

Mean score in eating disorders characteristics did not differ statistically significant between the intervention and the comparison group along the assessment times (no interaction group × time). Nevertheless, a considerable within-group difference was found between study times in the intervention (*p* = 0.004, W = 0.33) with moderate effect size. To better understand the impact of the program delivery or non-delivery on participants EAT-26 scores, differences between groups regarding the number of participants with pathological EAT-26 scores (>20) were performed. There was a statistically significant (*p* < 0.05) time effect in both the comparison group and the facilitators group. The of number of participants with pathological scores in the facilitators’ group decreased while increased in the comparison group, but no statistically significant group effect or group × time effect were found (Table 7).

## 4. Discussion

The current study describes the quantitative personal changes that occurred among students who facilitated a school-based wellness program. Facilitators were third-year college students in emerging adulthood. The hypothesis that delivering Young Favoring Myself will enhance self-esteem, self-care, body esteem, and media literacy compared to no change in the comparison group was only partially confirmed.

Participants in the facilitators’ group demonstrated statistically significant superiority, with a large effect size regarding the improvement in identifying advertisements strategies, body esteem related to weight, and the reduced impact of media messages pressure, although with a small effect size. Media literacy is a well-known mediator to improve body-esteem, which is often targeted in prevention programs. The statistically significant decrease in facilitators’ pressure due to media messages and the increase in participants’ identification of advertisement strategies may explain at least part of the improvement in body esteem. This connection has been previously reported [21,27].

Moreover, the number of participants whose EAT-26 scores showed pathological level (>20) decreased in the facilitators’ group from 5 to 2 while in the comparison group, it increased from 5 to 6 (no statistical significance). The superiority of those delivering a wellness program may be related to improved media literacy and body esteem. The superiority of the prevention program facilitation may be related to the intervention group’s participants experiential and empowering engagement, which fulfil students’ sense of belonging (to group leaders’ team), autonomy (facilitating group), and connectedness (to their personal self-care and to the faculty prevention team). These values were reported in students written reflections and suggested by the self-determination theory as major motivational values [24].

Self-esteem, total body esteem, self-care, and identification of advertisements strategies, were improved significantly in both groups along the time. Increased self-esteem is frequently reported in mentoring and peer-teaching [32,33]. The deeper information processing may enhance conceptual and practical learning and higher-order learning skills, such as collaboration, agency, critical thinking, metacognition, and regulation [34].

The improvement in both groups regarding some of the outcomes suggests that the chosen population–third-year students, might impose a selective bias. The academic platform exposes both the comparison and the intervention participants to various challenges that might impact the study outcomes. Opportunities to feel a sense of achievement, overcoming classroom discipline issues, modeling positive self-esteem, and creating a personalized version of the materials while receiving professional feedback may all impact.

Hence, the superiority of the intervention group should be interpreted with caution, and it cannot be concluded that the achieved improvements are due to the specific program [35].

To the best of our knowledge, no publications report on body esteem, media literacy, self-care, and eating disorders characteristics of prevention program facilitators, to be compared [21,36,37,38].

In line with others’ reports about the impact of a prevention program on the target population (not facilitators), only small effect sizes were achieved among program participants’ improvement (*p* = 0.06, Cohen’s D = 0.29) [10,39,40]. However, others reported that the prevention program’s effects were primarily short-term and diminished during follow-up [39,40,41]. In the current study, most improved outcomes were maintained at least three months after program conclusion. It can be speculated that the long-term efforts that the facilitators invested in delivering the program and the extended supervision they were provided both individually and in groups exerted an impact on them, and thus, continued months after program termination. The excellent adherence of participants to the intervention and the research process is also worthy of note. Other studies assessing outcomes among adolescents also reported maintained results in the long term [42,43,44,45].

Several limitations in this study should be noted. The main limitation is the lack of randomization within the controlled design. Although the comparison group participants represented a similar population in sociodemographic measures, they were not chosen in the same way as the intervention group, and confounding measures such as leadership, interpersonal, life skills, delivering other school-based activity, and other academic achievements were not explored. Since both groups represent a selective sample of primarily female college students, the generalizability of our findings is limited. Moreover, outcome data were collected based on self-report surveys, due to which potential self-report biases could not be ruled out [46,47]. To reduce biases, participants did not complete questionnaires during class but were provided online links, ensuring confidentiality.

Though all facilitators were committed to non-biased delivery of the program, and delivery fidelity was ensured by recording and supervising some sessions, conflict may arise from the requirement to deliver an intervention in an unbiased fashion while holding personal views about the research questions. This could impact the intervention’s delivery fidelity [48].

Finally, though most improvements among facilitators were maintained three months after the program’s conclusion, a longer follow-up assessment and larger sample size are warranted for future studies. Future studies may also consider testing other characteristics that may mediate the program’s impact.

## 5. Conclusions

This study indicates the potential contribution of delivering a prevention program on facilitators’ self-esteem, body-esteem, media literacy, and eating disorders’ characteristics. These outcomes are important protective factors during emerging adulthood. Moreover, supporting students with an adequate supervision system enhancing their sense of autonomy, competence, and relatedness while they facilitate prevention program, increases their internal motivation. Our findings may be helpful for enhancing student facilitator recruitment and commitment and scale up this prevention program. It also provides insight into a previously unexplored group of emerging adults that facilitates prevention and intervention programs. Offering these programs in an educational context could be valuable for accessing many integrators using fewer resources [49] while targeting self-care, self-esteem, and body satisfaction which are essential characteristics in contemporary society, especially among emerging adults.

## Figures and Tables

**Figure 1 ijerph-19-04278-f001:**
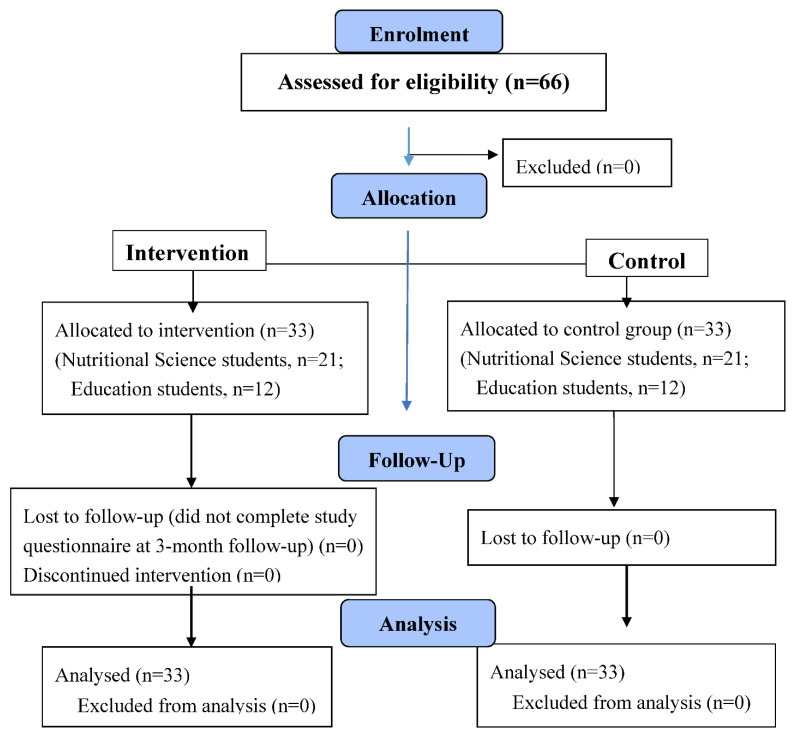
CONSORT 2010 Flow Diagram.

**Table 1 ijerph-19-04278-t001:** Content and description of the program sessions.

Session Number	Topic	Description
1	Introduction	Introducing the program’s objectives and goals, forming the groups, discussing expectations, and establishing the group contract.
2	Self-care	Discussing sleeping and eating hygiene, exercise, sharing self-care experiences and outcomes.
3	Self-preservation	Discussing territorial issues, self-space, and territorial boundaries.
4	Media literacy	Exploring and recognizing advertisements’ tactics and their impact on ourselves and discussing ways to address media temptations.
5	Our feelings	Facilitating a “feeling differentiation” activity. Sharing and recognizing observed and hidden feelings. Discussing feelings management strategies.
6	Accepting appearance differences	Discussing the ideal appearance in comparison to their unrealistic and narrow construction. Learning strategies to avoid and challenge comparisons and that different look is not necessarily a bad thing.
7	Accepting our weaknesses	Discussing how people turn their defects into productive effects. Learning how to accept disadvantages or weaknesses that cannot be changed.
8	My body and I	Exploring the physical changes during adolescence and role-play management strategies.
9	Adolescence rights and responsibilities	Discussing how growing up makes us take different responsibilities and gaining more independence and rights.
10	Summary and commitment	Reviewing key messages. Committing to engaging in positive self-care and positive body-image behaviors and rejecting risk factors.

**Table 2 ijerph-19-04278-t002:** Participants’ baseline demographic characteristics.

		Comparison n = 33	Intervention n = 33	*p*-Value ^1^
Age, mean ± standard deviation		26.3 ± 1.4	26.2 ± 1.9	0.35
Gender, n (%)	Male	5 (15.1)	5 (15.1)	1.0000
	Female	28 (84.9)	28 (84.9)	
Parental status, n (%)	Living alone	6 (18.2)	5 (15.1)	0.74
	Living in a relationship	27 (81.8)	28 (84.9)	
Socioeconomic status ^2^, mean ± standard deviation, median		0.9 ± 0.4, 0.8	0.9 ± 0.2, 1.0	0.74

^1^ Chi-square test for gender/parental status, Wilcoxon Two-Sample Test for age/Socioeconomic status; ^2^ Calculated by the number of people per room in residence.

**Table 3 ijerph-19-04278-t003:** Baseline differences between facilitators and comparisons in parametric outcome measures.

	Comparison Group n = 33		Intervention Group n = 33		*p* Value	Cohen’s d ^1^
	Mean	Std	Mean	Std		
Rosenberg	29.82	4.80	32.36	3.86	0.0208 *	0.58
Body-esteem Total	2.15	0.72	2.46	0.65	0.0736	0.45
Body-esteem Appearance	2.24	0.78	2.61	0.72	0.0483 *	0.49
Body-esteem Weight	2.38	0.87	2.37	0.80	0.9863	0.01
Body-esteem Attribution	2.08	0.79	2.45	0.64	0.0404 *	0.51
SATAQ-4 media	2.98	1.10	3.30	1.16	0.2690	0.28
Self-care	38.67	3.47	39.00	3.93	0.7161	0.09

* Significant differences; ^1^ Cohen’s d effect size interpretation: <0.20 small effect; >0.50 medium effect; >0.80 large effect.

**Table 4 ijerph-19-04278-t004:** Baseline differences between facilitators and comparisons in non-parametric outcome measures.

	Comparison Group n = 33			Intervention Group n = 33			*p* Value (Effect Size ^1^)
	Mean	Std	Median	Mean	Std	Median	
ADVER	4.33	1.73	4.00	4.67	1.43	5.00	0.4645 (0.090)
EAT_26	10.00	11.40	6.00	11.82	8.05	10.00	0.0522 (0.239)

^1^ Wilcoxon effect size (r) interpretation: <0.3 small; <0.5 moderate; >0.5 large effect; EAT-The Eating Attitudes Test.

**Table 5 ijerph-19-04278-t005:** Differences in outcome measures between groups and changes within each group (parametric outcome measures).

		Comparison Group			Intervention Group			Effect	F (df)	Effect Size Marginal R^2^
		Baseline	Post-Intervention	Follow-Up	Baseline	Post-Intervention	Follow-Up			
Rosenberg ^1^	Mean	29.82	30.08	31.4	32.36	33.42	33.88	Group	31.81 (1.63) ***	0.072
	Std	4.80	4.78	5.14	3.86	4.27	4.34	Time	4.94 (2.126) **	
	n	33	33	33	33	33	33	Group × Time	0.48 (2.126)	
Body-esteem Total	Mean	2.15	2.25	2.39	2.46	2.63	2.65	Group	4.18 (1.64) *	0.066
	Std	0.72	0.70	0.75	0.65	0.65	0.63	Time	6.61 (2.128) **	
	n	33	33	33	33	33	33	Group × Time	0.51 (2.128)	
Body-esteem Appearance ^1^	Mean	2.24	2.39	2.17	2.61	2.72	2.61	Group	0.60 (1.63)	0.088
	Std	0.78	0.72	0.94	0.72	0.69	0.78	Time	6.13 (2.126) **	
	n	33	33	33	33	33	33	Group × Time	0.36 (2.128)	
Body-esteem Weight	Mean	2.38	2.17	2.26	2.37	2.51	2.57	Group	1.25 (1.64)	0.025
	Std	0.87	0.89	1.01	0.80	0.83	0.79	Time	0.53 (2.128)	
	n	33	33	33	33	33	33	Group × Time	3.28 (2.128) *	
Body-esteem Attribution	Mean	2.08	2.02	2.31	2.45	2.62	2.68	Group	8.60 (1.64) **	0.115
	Std	0.79	0.77	0.59	0.64	0.65	0.69	Time	6.13 (2.128) **	
	n	33	33	33	33	33	33	Group × Time	1.98 (2.128)	
SATAQ-4 media	Mean	2.98	3.20	3.11	3.30	2.95	3.08	Group	0.00 (1.64)	0.011
	Std	1.10	1.21	1.13	1.16	1.02	1.14	Time	0.17 (2.128)	
	n	33	33	33	33	33	33	Group × Time	3.28 (2.128) *	
Self-care	Mean	38.67	41.15	39.70	39.00	41.06	39.76	Group	0.03 (1.64)	0.085
	Std	3.47	2.87	3.04	3.93	2.66	2.36	Time	16.17 (2.128) ***	
	n	33	33	33	33	33	33	Group × Time	0.14 (2.128)	

^1^ Baseline scores for these variables were included in the repeated-measures ANOVA analysis as covariates; * *p* < 0.05 ** *p* < 0.01 *** *p* < 0.001.

**Table 6 ijerph-19-04278-t006:** Differences in outcome measures between groups and changes within each group (non-parametric outcome measures).

		Comparison Group n = 33				Intervention Group n = 33				*p* Value ^1^ (Effect Size)
		Mean	Std	Median	*p* Value ^2^ (W)	Mean	Std	Median	*p* Value ^2^ (W)	
ADVER	Baseline	4.33	1.73	4.00	0.0020 (0.38)	4.67	1.43	5.00	<0.0001 (0.57)	0.4645 (0.090)
	Post-intervention	4.30	1.63	5.00		5.64	1.54	6.00		0.0005 (0.431)
	Follow-up	5.12	1.19	5.00		5.79	1.49	6.00		0.0255 (0.275)
EAT-26	Baseline	10.00	11.40	6.00	0.0780 (0.15)	11.82	8.05	10.00	0.0041 (0.33)	0.0522 (0.239)
	Post-intervention	7.36	8.46	4.00		9.30	7.87	8.00		0.0656 (0.227)
	Follow-up	9.94	9.60	5.00		10.00	9.38	8.00		0.5586 (0.072)

^1^ Group effect (Wilcoxon Two-Sample Test). Wilcoxon effect size (r) interpretation: <0.3 small; <0.5 moderate; >0.5 large effect. ^2^ Time effect (Friedman’s Chi-Square Test). Kendall’s W effect size interpretation: <0.3(small effect), <0.5 moderate and >0.5 (large).

**Table 7 ijerph-19-04278-t007:** Differences between the number of participants with eating disorders characteristics.

		Comparison Group n = 33		Intervention Group n = 33		Effect	Chi-Square (df)	Effect Size
		Score > 20						
		n	%	n	%			
EAT-26	Baseline	5	15.15	5	15.15	Group	0.63 (1)	0.027
	Post-intervention	3	9.09	2	6.06	Time	7.17 (2) *	
	Follow-up	6	18.18	2	6.06	Group × Time	3.16 (2)	

Within each group, along with the assessment times. * *p* < 0.05; EAT-26: The Eating Attitudes Test.

## Data Availability

The data that support the findings of this study are available from the first author upon reasonable request and with the permission of The College IRB. Restrictions apply to the availability of these data by the ministry of education, and so are not publicly available.

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
