# Peer review of "The Impact of Delivering School-Based Wellness Programs for Emerging Adult Facilitators—A Quasi-Controlled Clinical Trial"

_ijerph, 2022, doi:10.3390/ijerph19074278_

Round 1

Reviewer 1 Report

The review comments are listed as follows.

  1. About the 10 topics and the corresponding content in the "Favoring Myself" preventative program, the authors should have some descriptions why they should have those 10 topics and the corresponding content design in the program.
  2. The authors should explain why their study focuses on the self-esteem, self-care, media literacy, and body esteem in the school-based preventative program.
  3. The “Twenty-four” in line 133 can be expressed by a number form, i.e. 24.
  4. The quality of Figure1 is not very well; it had better for the authors to redraw Figure1. Moreover, the authors should have more descriptions about the flow diagram shown in Figure 1.
  5. The authors should further explain the association between the study population and Figure 1.
  6. Line 160 mentioned “a computerized self-administered questionnaire (Supplement 2)”; however, Supplement 2 cannot be found in the manuscript.
  7. Some abbreviations, such as STD in Table 1 and EAT in Table 7, should have the full text in the list of abbreviations.
  8. Several questionnaires were used in this study; it had better for the authors to list all the questions or summarize those questions in the questionnaire and the group semi-structured interviews so readers could understand the contents of the questionnaire and the group semi-structured interviews.
  9. In line 241, the “Body Esteem Appearance” should be expressed as “Body Esteem Appearance (p=0.040)”.
  10. In line 269, the “up assessments (p=0.025, W=0.275)” should be “up assessments (p=0.025, W=0.275).”.
  11. In Section 3, the authors described the study results with the “facilitators and comparison group” or the “the intervention and the comparison groups”; however, examining Table 3 ~ 7, the “Control group” and “Intervention group” are used in those tables to show the statistical results. It had better for the authors to use the same terms in descriptive texts and tables.
  12. In line 291, the font of “Table 6.” is different from the font of the others tables’ titles.
  13. The editing format of line 378 should be same as the other lines in the “List of abbreviations”.
  14. 49 papers are listed in “References” in the manuscript; however, only 48 papers are cited. The authors should check the literature citations in the manuscript.

Author Response

1+2. To address the reviewer comment we added in the revised manuscript line 126 the two reasons for the inclusion of this topics: a) these topics are considered as protective factors against risk behaviors b) they are evidence-based intervention's topics

3. Line 133-corrected

4. As the reviewer suggested, Figure 1 was improved

5. We added an explanation on line 137 - "All eligible participants completed all assessments' questionnaires".   

6. The reviewer is right. We deleted the supplement two since the questionnaire is in Hebrew 

7. Comment about the abbreviations-corrected in the revised manuscript only in the Tables the reviewers mentioned. In the other tables we can not use the full name of STD and thus there is a list of abbreviations  after the conclusions.

8. We added supplement 1 with sample questions used in the semi-structured interviews line 602

9. Corrected. 

10. corrected

11. As the reviewer suggested, all terms in the texts and table are similar now. 

12. Fonts are the journal's .

13. Journals fonts

14. Both are 49

This manuscript present only the quantitative data. Thus the qualitative methods and the promised appendix were deleted from the revised manuscript. 

Reviewer 2 Report

The research is oriented to the study of a relevant topic for the training of higher education students. According to the structure of the manuscript: (1) The research problem is clearly described, which allows a coherent construction of the object of study, based on an adequate review of empirical background. However, it is suggested to conceptually and theoretically describe the study variables (self-esteem, body-image, and media literacy), because there are different perspectives of well-being. For this reason, it is suggested to specify the theoretical-conceptual scope that guides the observation of the study variables, which will contribute to strengthening the development of the conclusions. (2) The references used are relevant and current. (3) An adequate methodological strategy is proposed to achieve the stated objectives. (4) It is important to improve the conclusion since a greater articulation could be made with theoretical and empirical arguments that allow supporting the research findings. Also, it is important to describe the practical implications of the findings related to the implementation of wellness programs.

Author Response

  1. As the reviewer suggested, the theoretical background of the program's topics and the study hypothesis were added- line 87 in the revised manuscript:  "The program’s topics targets protecting factors (knowledge and understanding of self care in various situations, self-esteem, positive body esteem, media literacy, positive defense mechanisms). The study hypothesis relied on Deci and Ryan’s self-determination theory (24) assuming that when students achieve sense of autonomy, relatedness and competence through the support system provided while facilitating prevention program and through the emphasize on certain topics and assignments, they are highly motivated and personally impacted. 
  2. We also added in the discussion "

    The superiority of the prevention program facilitation may be related to the intervention group’s participants experiential and empowering engagement ,which fulfil students’ sense of belonging (to group leaders’ team), autonomy (facilitating group) and connectedness (to their personal self-care and to the faculty prevention team). These values were reported in students written reflections  and suggested by the self-determination theory as major motivational values [24]."

  3. The practical implications were in the conclusions were improved with the addition of : "These outcomes are important protective factors during emerging adulthood. Moreover, supporting students with adequate supervision system enhancing their sense of autonomy competence and relatedness while they facilitate prevention program, increase their internal motivation. Our findings may be helpful to enhance student facilitator recruitment and commitment and scale up this prevention program."

Reviewer 3 Report

Providing tables and figures can be extremely valuable to a manuscript as they provide a visual representation of a study and findings. It there is a way to insert text between tables to help the reader transition from one to the other, that would be very helpful. 

Additional Comments:

It is not clear if the layout of figure 1 (flow diagram) got altered in the submission process!  If not, it could be restructured. 
It would help the reader if the "Data Collection Procedure" is further explained. 
This is a minor formatting issue: the alignment within some tables could be improved. 
The chosen elements for the study are listed in Table 1: It would be helpful to the reader to have a brief explanation of the choices and the reasons behind these choices. This could be accomplished through a brief narrative addressing the steps that led to these choices. 

Author Response

The chosen elements are considered protective factors against risk behaviors as well as evidence-based prevention objectives that were improved by school-based prevention program - Added in the revised manuscript